# Evaluation of Genomic Prediction for Pasmo Resistance in Flax

**DOI:** 10.3390/ijms20020359

**Published:** 2019-01-16

**Authors:** Liqiang He, Jin Xiao, Khalid Y. Rashid, Gaofeng Jia, Pingchuan Li, Zhen Yao, Xiue Wang, Sylvie Cloutier, Frank M. You

**Affiliations:** 1Ottawa Research and Development Centre, Agriculture and Agri-Food Canada, Ottawa, ON K1A 0C6, Canada; liqiang.he@canada.ca; 2State Key Laboratory of Crop Genetics and Germplasm Enhancement, College of Agriculture, Nanjing Agricultural University/JiangSu Collaborative Innovation Center for Modern Crop Production, Nanjing 210095, China; xiaojin@njau.edu.cn (J.X.); xiuew@njau.edu.cn (X.W.); 3Morden Research and Development Centre, Agriculture and Agri-Food Canada, Morden, MB R6M 1Y5, Canada; khalid.rashid@canada.ca (K.Y.R.); lipingchuan@gmail.com (P.L.); zhen.yao@canada.ca (Z.Y.); 4Crop Development Centre, University of Saskatchewan, Saskatoon, SK S7N 5A8, Canada; gaofeng.jia@usask.ca

**Keywords:** genomic selection, genomic prediction, genotyping by sequencing, pasmo resistance, pasmo severity, quantitative trait loci, single nucleotide polymorphism, *Septoria linicola*, flax

## Abstract

Pasmo (*Septoria linicola*) is a fungal disease causing major losses in seed yield and quality and stem fibre quality in flax. Pasmo resistance (PR) is quantitative and has low heritability. To improve PR breeding efficiency, the accuracy of genomic prediction (GP) was evaluated using a diverse worldwide core collection of 370 accessions. Four marker sets, including three defined by 500, 134 and 67 previously identified quantitative trait loci (QTL) and one of 52,347 PR-correlated genome-wide single nucleotide polymorphisms, were used to build ridge regression best linear unbiased prediction (RR-BLUP) models using pasmo severity (PS) data collected from field experiments performed during five consecutive years. With five-fold random cross-validation, GP accuracy as high as 0.92 was obtained from the models using the 500 QTL when the average PS was used as the training dataset. GP accuracy increased with training population size, reaching values >0.9 with training population size greater than 185. Linear regression of the observed PS with the number of positive-effect QTL in accessions provided an alternative GP approach with an accuracy of 0.86. The results demonstrate the GP models based on marker information from all identified QTL and the 5-year PS average is highly effective for PR prediction.

## 1. Introduction

Flax *(Linum usitatissimum* L.) is an important food and fibre crop cultivated and grown in cooler regions of the world, such as Canada [1]. Pasmo, elicited by the fungus *Septoria linicola*, is one of the most widespread diseases of flax, causing reductions in seed and oil yield, as well as fibre quality and durability [2]. Developing resistant cultivars is the most viable and effective option to control this disease that has become widespread in all flax production areas of North America and other parts of the world. Resistance to pasmo has a low heritability [3] and is quantitatively inherited [4]. Large variations in pasmo disease severity were observed in the flax core collection, which can be capitalized upon to develop resistant cultivars [3]. Phenotypic recurrent selection is typically used to develop cultivars with improved resistance and selection is usually carried out based on phenotypic assessments of resistance in field conditions [5]. However, field assessment of pasmo severity (PS) in germplasm and breeding lines is costly and, is heavily influenced by the environments due to strong genotype × environment (G × E) interactions [3,4].

With the advancements in molecular marker development over the last decade, efforts to use marker-assisted breeding strategies have been pursued. One such strategy involves identifying quantitative trait loci (QTL) in biparental mapping populations and using markers to efficiently backcross QTL into elite breeding materials [6]. This so-called marker-assisted recurrent selection (MARS) or simply marker-assisted selection (MAS) characterizes many breeding programs that employ molecular markers to select non-phenotyped individuals for crossing and downstream selection of segregating populations [7]. This method is suitable for the selection of monogenic or oligo-genic architectures but has limited use for quantitative traits controlled by many genes of smaller effects [8]. Genomic selection (GS) or prediction (GP) is an alternative marker-assisted breeding strategy better suited to polygenic quantitative traits, especially those with low heritability, because it makes use of all marker effects across the entire genome to calculate genomic estimated breeding values (GEBVs) [9] for individual plant selection [9,10].

In GP, a training population (TP) is genotyped with genome-wide markers and phenotyped for the trait(s) under selection; statistical models that best predict the breeding values from the marker data are then applied to select non-phenotyped germplasm. GP has been used to select for disease resistance in several crops such as *Fusarium* head blight (FHB) in wheat, a typically quantitatively inherited trait with predominantly additive genetic variation, where GP had a significantly higher accuracy than pedigree-based information alone [11]. GP feasibility has also been studied for selection of wheat rust resistance and was found particularly effective when validation lines had at least one which is close to the reference lines [12]. The implementation of GP on northern leaf blight, a complex genetic architecture trait in maize, resulted in superior gains and reduced breeding cycle time to ≤80% of the phenotypic cycle [13]. Despite the many successful examples, the use of GP to improve disease resistance in crops has been challenging for two reasons: (i) selection for major resistance genes can be ephemeral due to changes in pathogen races; and (ii) breeding for minor resistance genes with small effects may face the remarkable complexities encountered in GP [14].

The fast-evolving genotyping platforms have been a game-changer in the implementation of GP, allowing the production of large numbers of genome-wide markers, whereas progresses in phenotyping were not associated with similar cost reduction or quantum leaps in throughput. Given the number of markers (*p*) and sample size (*n*) in a given population, there are many more *p* effects to be estimated than the *n*, leading to an infinite number of possible marker effect estimates [15], that is, the so-called “large *p*, small *n* problem” (*p* >> *n*) when applying markers to predict phenotypes [11]. Several GP statistical models have been proposed to address this issue [16]. For example, the ridge-regression best linear unbiased prediction (RR-BLUP) is a mixed linear model that considers markers as random effects. Covariance between markers is considered to be zero and the marker variance is assumed to be the total genetic variance divided by the number of markers. The variance is assumed to be equal for all markers, allowing many more marker effects to be estimated than there are phenotypic records [17]. Unlike RR-BLUP, the Bayesian LASSO (BL) assumes markers to have unequal variances and, performs continuous shrinkage and variable selection simultaneously, with small-effect markers shrinking more severely than larger-effect loci. In the *p* >> *n* setting, LASSO will select at most *n* − 1 variables and set the effects of the remaining predictors at zero [18]. Although the problem is solved statistically in these models, improving the accuracy and efficiency of GP by reducing the number of genome-wide markers would be advantageous because any increment in the TP size comes at a cost [19,20,21,22]. Genome-wide association study (GWAS) is an approach to identify genome-wide markers linked to QTL, resulting in a limited number of favourable genetic loci responsible for traits of interest [23]. For example, GP of crown rust resistance in *Lolium perenne* demonstrated GWAS’s ability to identify and rank markers, which enabled the identification of a small subset of single nucleotide polymorphisms (SNPs) that could achieve predictive abilities close to that attained using the complete marker set [24]. Utilization of GWAS removes a large proportion of unrelated markers and in the construction of prediction models.

The only GP empirical study published to date in flax, which used bi-parental populations for yield, oil content and fatty acid composition traits, indicated that GP could increase genetic gain per unit time in linseed breeding. The GP results significantly exceeded those from direct phenotypic selection, especially for traits with low broad-sense heritability [25]. Resistance to flax pasmo is polygenic. Our previous study reported 500 non-redundant QTL for PR from 370 diverse flax accessions of a core collection based on five-year pasmo field assessments; of those, 134 QTL were statistically stable in all five years and 67 had relatively stable and large effects [4].

The objective of this study was to evaluate the potential of QTL markers in GP and compare the GP efficiency affected by different markers, including genome-wide SNPs and QTL markers, to provide a realistic and highly accurate model for germplasm evaluation and parent selection in pasmo resistance breeding.

## 2. Results

### 2.1. Evaluation of Pasmo Resistance

PS ratings at green boll stage or maturity across five consecutive years were similar but on average PS ratings in 2014 and 2016 were higher than those in other years (Table 1). They had single peak distributions but skewed towards high PS ratings except for those in 2014 (Figure 1). Scatter plots of PS ratings between years indicated strong genotype × year interaction even though statistically significant correlations of PS ratings between years were observed (Figure 1), as shown in the variance analysis results in the previous study [4]. However, the Pearson correlations of 5-year averages of PS ratings (PS-mean) with those in individual years (*r* = 0.72–0.83) were much higher than the Pearson correlations between individual years (*r* = 0.31–0.62) (Figure 1), implying that the mean PS ratings over multiple years or environments were a more suitable data set than individual year’s data sets for model construction of genomic prediction.

### 2.2. Evaluation of Marker Sets Used in Genomic Prediction

Four marker sets were used for GP of pasmo resistance. The first marker set contained 52,347 genome-wide SNPs (SNP-52347) that were correlated to the five-year average PS and the PS of the five individual years at a 10^−5^ probability level [4]. The other three marker sets were the 500 unique QTL (SNP-500QTL), the 134 QTL statistically stable over five consecutive years (SNP-134QTL) and the 67 stable and relatively large-effect QTL (SNP-67QTL) sets previously identified [4]. The SNP-500QTL dataset comprises markers for all small- or large-effects, including QTL stable across environments and environment-specific QTL identified using three single-locus and seven multi-locus statistical models and all six phenotypic datasets (Figure 2). The SNP-134QTL dataset is a subset of the SNP-500QTL dataset whereas SNP-67QTL is a subset of the former; all SNP-500QTL markers were included in SNP-52347. These four marker sets explained 54%, 72%, 27% and 29% of the phenotypic variation of the five-year PS average (PS-mean), respectively; these values exceeded those of the individual year PS data (Table 2). Although SNP-500QTL was a subset of SNP-52347, this marker set explained a greater percentage of the phenotypic variation for PS than SNP-52347 for all datasets.

### 2.3. Accuracy of Genomic Prediction in Relation to Marker Sets and Pasmo Severity Datasets

Genomic prediction models were constructed using RR-BLUP with pairwise combinations of the four marker sets and the six PS datasets. Statistical models for the 24 combinations were generated and evaluated for their accuracy (*r*) and relative efficiency (*RE*) using a five-fold random cross-validation scheme (Table 3). *RE* represents the relative efficiency of GP over direct phenotypic selection which depends on the heritability of a selective trait. Direct phenotypic selection for a trait was considered to have a baseline efficiency of 1. Thus, *RE* values greater than 1 indicate GP models more efficient than direct phenotypic selection in one selection cycle [25,26,27]. Analysis of variance (ANOVA) (Appendix A) indicated that *r* and *RE* both significantly differed among the four marker sets and the six PS datasets; there was also a significant interaction effect between marker sets and PS datasets (Appendix A). Owing to the significant marker × phenotype dataset interaction, multiple comparisons of the 24 combinations were performed. For all marker sets, the PS-mean models significantly outperformed those based on individual year datasets (Table 3). The SNP-500QTL marker set models generated significantly higher *r* and *RE* values than any other marker sets (Figure 3). Interestingly, the SNP-67QTL derived models produced slightly but significantly higher values of *r* and *RE* than SNP-134QTL models. The highest *r* and *RE* values were obtained for models combining the SNP-500QTL and PS-mean datasets (Table 3, Figure 3). Intriguingly, the SNP-52347 models yielded the lowest *r* and *RE* values despite including all QTL markers (Table 3, Figure 3); both BL and Bayesian ridge regression (BRR) corroborated this finding (Appendix A). No significant differences in *r* and *RE* values were observed among the three statistical models: RR-BLUP, BL and BRR (Appendix A).

### 2.4. Sample Size of Training Populations versus Genomic Prediction Accuracy

To find an optimal size for the TP, the relationship between TP size and prediction accuracy was analysed. TPs of various sizes from 18 to 351, corresponding to 5% to 95% of the total 370 accessions, were used to build models with the SNP-500QTL marker set and the PS-mean phenotypic dataset. The prediction accuracy significantly increased for TP sizes up to 100, followed by smaller accuracy gains with every additional TP size increments (Figure 4). A GP accuracy >0.9 was obtained once the TP size reached 185 (Figure 4).

### 2.5. Prediction Models of Pasmo Resistance

All 370 accessions were used as a training population to build a prediction model using the SNP-500QTL genotypic dataset and the PS-mean phenotypic dataset because this combination outperformed all other models. The model was then employed to predict PS in each year (Table 4). Prediction accuracies (*r*) ranging from 0.71 to 0.81 and *RE* values of 1.42 to 1.62 were obtained when predicting PS for individual years (Table 4).

A prediction accuracy as high as 0.98 and a *RE* value of 1.96 were obtained when the model was used to predict PS-means of the 370 accessions (Table 4). A linear relationship was observed between the observed (*y*) and predicted PS (*x*): *y* = 1.0522*x* − 0.3267 (*R*^2^ = 0.96) (Figure 5a). Based on this equation, the average prediction interval between the two red dashed lines, representing the 95% confidence interval, was only less than 1 (an average of 0.97) on the PS ratings (Figure 5a).

NPQTL in the 370 accessions for the 500 QTL set was tallied. Significant linear correlation between PS-mean and NPQTL (*r* = 0.86 or *R*^2^ = 0.73) was observed (Figure 5b). This correlation was less than but close to the accuracy of the GP model with SNP-500QTL and higher than the GP models using other marker sets (Table 3). However, the single linear regression equation (*y* = −0.0262*x* + 11.934) of the observed PS (*y*) to NPQTL (*x*) had a large standard deviation for each prediction value, with an average prediction interval width of 2.70, nearly three times the average prediction interval width of the GP model; that is, the NPQTL model had a higher prediction error than the GP model.

### 2.6. A Case Study of Genomic Prediction

To assess GP prediction accuracy, a training-testing partition was generated with random assignment of breeding lines to either training or testing subsets. Considering the different improvement status of accessions in the population (cultivars, breeding lines, landraces or unknown types) and different levels of resistance, we randomly chose 20% of the 370 accessions in the population, that is, 93 accessions (52 cultivars, 21 breeding lines, 3 landraces and 17 unknown types) as validation dataset, that is, a five-fold random cross-validation set. To predict the PS of these 93 accessions, a RR-BLUP model using the SNP-500QTL set and the PS-mean of the remaining 277 accessions as TP set was built to predict PS. The predicted results are shown in Figure 5c and Appendix A. The prediction accuracy was as high as 0.95 (*r* between observed and predicted PS). Similarly, a linear regression model of observed PS (*y*) to NPQTL (*x*) of the 277 accessions (the same TP as GP) produced *y* = −0.026*x* + 11.902 (Appendix A), which was similar to the regression equation previously obtained with the complete accession set (Figure 5b). Using this prediction model, predicted PS and intervals were calculated (Figure 5d, Appendix A). The prediction accuracy of 0.92 for NPQTL was slightly inferior to that of the GP model. The observed PS values all fell within prediction intervals (Appendix A).

## 3. Discussion

Cross-validation remains the most popular method to evaluate GP accuracy [14,28]. Our RR-BLUP model prediction accuracy of 0.92 for PR is the highest of all published GP models for plant disease resistance traits [14]. This model is especially valuable because PR has low heritability and high inheritance complexity [3,4]. The QTL markers, multi-year phenotypic data and the genetic diversity and size of the population likely contributed positively to this high prediction accuracy [29].

### 3.1. All Detected QTL Used as Markers in Genomic Prediction 

Three sets of QTL markers (SNP-500QTL, SNP-134QTL and SNP-67QTL) and a genome-wide SNP marker set (SNP-52347) were evaluated here. GP models built using SNP-500QTL consistently outperformed models derived with any of the other three marker sets (Table 3, Appendix A), lending credence to the robustness and reliability of the QTL identified using multiple single-locus and multi-locus GWAS statistical methods [4]. Most GWAS aim to detect large-effect QTL, such as the SNP-67QTL set. While potentially useful in MAS, these tend to explain a reduced portion of the phenotypic variation compared to more comprehensive models (Table 2). Consequently, the GP models built with such marker sets have lower GP accuracies. Therefore, using all potential QTL associated with the selective trait to build GP models is advantageous because it greatly improves prediction accuracy. Prediction accuracies of models obtained with SNP-134QTL and SNP-67QTL data sets were comparable (Table 3, Appendix A) and they explained a similar proportion of the phenotypic variation for PS (Table 2), confirming the redundancy or overlap between the two datasets. Removal of redundant QTL from SNP-134QTL to produce SNP-67QTL produced slightly higher accuracy models (Figure 3). Simplifying GP models by removal of redundant and unrelated markers will ease the practical implementation of GP in breeding programs.

### 3.2. Superior Performance of Genomic Prediction Combined with GWAS

Surprisingly, the GP models built using SNP-52347 generated a lower prediction accuracy than the models with SNP-500QTL (Table 3, Appendix A), regardless of the statistical methods (Appendix A). Similarly, SNP-52347 explained a lower percentage of the phenotypic variation for PS than SNP-500QTL (Table 2). Besides interaction between SNPs, introduction of noise from genome-wide markers [30], the low prediction accuracy may also be owing to some of the erroneously called SNPs and imputation of missing SNP data. SNP-500QTL includes all or nearly all QTL potentially associated with PS; additional markers, not only failed to increase but actually reduced the prediction accuracy, further emphasizing the effectiveness of the QTL identification methodology adopted in our previously published GWAS study [4]. Similar findings were found for FHB in wheat where deoxynivalenol (DON) concentration QTL-linked markers significantly improve prediction accuracy compared to random genome-wide markers [30]. Markers linked to QTL underlying important traits are deemed more useful for prediction strategies because genome-wide markers may introduce noise, thereby reducing accuracy [30]. Using QTL for GP models may be beneficial to balance genetic backgrounds along with maximum gain of breeding value [31]. Genome-wide prediction models based on ~5000 SNPs from de novo GWAS for tropical rice improvement were as effective for prediction as the full marker set of 108,005 SNPs, indicating that the relationship between marker number and prediction accuracy is neither strict nor linear [32]. To sum up, combined applications of the QTL discovered via GWAS and the accelerated breeding cycles through GP facilitate the full use of genome-wide markers in crop disease resistance breeding [10,33]. Removal of redundant markers has the potential to alleviate the effect of the “large *p*, small *n*” issue.

### 3.3. Accuracy of GP Modelling by Environment, Training Population and Statistical Methods

G × E interactions, which affects the accuracy of trait assessment, are common for plant traits. A strong G × E interaction was observed in flax PR [4]. As a consequence, different PS QTL were identified for individual years and for the 5-year average [4]; similarly, GP efficiencies differed when individual yearly and average PS data sets were used as training sets (Table 3). The highest accuracies were obtained when the 5-year mean phenotypic data was used as training data (Table 4), suggesting that the average phenotypic data across multiple environments should be used for GP model construction. Because phenotypic values of genotypes in each year had one replication, the average phenotypic data across multiple years is actually equivalent to the best linear unbiased prediction values (BLUPs) or the best linear unbiased estimators (BLUEs). Therefore, the means across multiple environments estimate or reflect the true breeding values of a trait.

Some studies report that prediction accuracy of GP is highly affected by the size of the TP. In general, the prediction accuracy increases with TP size [21,28,29,34,35,36]. In the GP of seed weight in soybean, for example, prediction accuracy was sensitive to changes in TP size, which may have led to changes of relatedness between training and validation sets [21]. Lorenzana and Bernardo observed that, in an Arabidopsis family, prediction accuracy improved by 0.10 when TP size increased from 48 to 96, by an additional 0.07 when TP size was increased to 192 and by a further 0.05 with a TP size of 332 [37]. Here GP accuracy >0.9 was observed when the TP size reached 185 which slightly increased to 0.921 with a TP size of 314 (Figure 4). Large TPs provide the statistical power needed to improve prediction accuracy [38], especially for traits with low heritability [34,39]. When TP size is sufficiently large, even low heritability traits can be accurately predicted [28,40], including the low heritability PS studied therein. Diversity of the population also affect prediction accuracy [21,29,34,41,42,43]. A diverse TP may contain more QTL associated with selective traits and increase the correlation of the TP with validation populations (VPs) or test/prediction populations (PPs), resulting in a subsequent increase in prediction accuracy. Although some breeding lines [11,30,44] and bi-parental derived lines [25,41,45,46] are used for TPs, many studies have opted for a more diverse TP germplasm [29,41,42,43]. Our core collection TP preserves the variation present in the world collection of 3378 accessions maintained by Plant Gene Resources of Canada (PGRC) and represents a broad range of geographical origins, different improvement statuses (landraces, historical and modern cultivars, breeding lines) and two morphotypes (linseed and fibre types) [1,3]. This collection also contains most parents of modern Canadian flax cultivars [25]. Therefore, diverse phenotypic and genetic variabilities within the flax core collection render it useful as a resource for breeding and as a TP for GP model construction.

A variety of statistical methods have been proposed to estimate marker effects for GP. In general, GP methods are based on additive genetic models and their accuracies may vary depending on genetic architecture of target traits. According to the assumptions for statistical distributions of the marker effects, two groups of GP models have been proposed. The first group of models, such as RR-BLUP, genomic BLUP (GBLUP) and BRR, assume that all markers have some effects on the target trait and the same variance, that is, all makers contribute to the variation of the trait. The second group of models, including BayesA, BayesB, BayesC and BL, assume a specific variance for each marker. Some of these models such as BayesB, BayesC and BL, also allow variable (marker) selection when some of markers have very small or no effects. Based on these assumptions, the first group of models are expected to be useful for complex quantitative traits that have a polygenic architecture, while the second group of models are suitable for traits that controlled by a small number of genes or QTL with large effects. Several studies have shown better performance of BayesB for traits controlled by a few of genes with large effect [47,48,49,50]. Some simulation studies have also shown that BayesB outperformed GBLUP that is equivalent to RR-BLUP, when the number of QTL underlying a trait are small [47,51]. However, BayesB, RR-BLUP and other models had a similar prediction accuracy under the infinitesimal model [51] or for some complex traits [19,49]. In this study, no difference among RR-BLUP, BRR and BL was observed (Appendix A), primarily because flax pasmo resistance is a complex and polygenic trait and most of QTL associated with it had similar and small effects (Figure 2). RR-BLUP is most commonly used because of some superior features [11,14,42,52,53,54]. For example, RR-BLUP successfully recognized complex patterns with additive effects and delivered good GP in wheat disease resistance [55]. RR-BLUP also has a clear-cut computational efficiency compared with any other statistical models [11,54,56,57]. Here the RR-BLUP model with the 500 QTL markers and the 5-year mean PS produced high prediction accuracy and is therefore recommended for the prediction of PR in flax.

### 3.4. Pasmo Severity Prediction Using Number of Positive-Effect QTL

A highly significant correlation (*r* = 0.86 or *R^2^* = 0.73) between NPQTL and PS (Figure 5b) provides an alternative approach to directly predict PS phenotypes. The prediction accuracy using the linear regression equation of PS to NPQTL was inferior to the GP model (Figure 5) because the QTL effects were variable (Figure 2), whereas the linear regression equation considered only the number of QTL but not their individual effects. However, NPQTL is advantageous because it can be readily calculated based on the genotyping by sequencing (GBS) or other genotyping data for the QTL markers [14] and the prediction accuracy based on the NPQTL is comparable to most GP models. Thus, the NPQTL-based prediction equation provides a simple alternative model for PS prediction.

### 3.5. Breeding Application of Genomic Prediction

Plant breeding is to pyramid favourite alleles from distinct parents using different approaches such as conventional crossing, mutation or transgenic methods to develop new varieties. However, most traits of agronomic importance are genetically controlled by polygenes and have a low heritability such as seed yield and horizontal resistance to diseases. Conventional phenotype selection for these traits is usually inefficient because assessment for them must be performed in multiple environments to obtain breeding values of individuals and thus it is very costly, time consuming and inaccurate; and also because of difficulty of evaluation in fields, greenhouses or laboratories. GS or GP provides an efficient approach to increase selection efficiency by not only increasing selection accuracy but also shortening breeding cycles [58]. In this study, we demonstrate a good example of GP for flax pasmo resistance that is environment-sensitive, costly and difficult for field evaluation. As high as 0.92 of prediction accuracy was obtained for PR, corresponding to 1.84 of relative efficiency over the direct phenotypic selection (Table 3), demonstrating efficiency of GP for low heritability traits. Because the training population underlying the GP models is a diverse germplasm collection that contains more than 90 breeding lines and 245 varieties from different breeding programs [3], the GP models developed in this study are expected to be used for germplasm evaluation, parent selection and individual selection of segregation populations for PR.

## 4. Materials and Methods

### 4.1. Population

A total of 370 diverse flax accessions from the core collection [1] were used to evaluate different GP models. This subset of the core collection collected from 38 countries in 12 geographic regions has been used to identify the QTL associated with PS used in our PS models [4].

### 4.2. Pasmo Resistance Data

All flax accessions were assessed for PS in the same pasmo nursery from 2012 to 2016 at the Morden Research and Development Centre, Agriculture and Agri-Food Canada (AAFC), Morden, Manitoba, Canada [4]. A type-2 modified augmented design (MAD2) [59,60] was used for the field trials [3]. Accessions were seeded during the second or third week of May every year. Approximately 200 g of pasmo-infested chopped straw from the previous growing season was spread between rows as inoculum when plants were approximately 30-cm tall. A misting system was operated for 5 min every half hour for 4 weeks, except on rainy days, to ensure conidia dispersal and disease infection and development. Field assessments were conducted at the early (P1) and late flowering stages (P2, 7–10 days after P1), the green boll stage (P3, 7–10 days after P2) and the early brown boll stage (P4, 7–10 days after P3). In 2014 and 2015, only the first three field assessments were conducted because early maturity of the plants did not allow for a fourth rating. The PS observed at green boll stage or maturity was used for GP as previously described [4]. PS was assessed on leaves and stems of all plants in a single row plot using a 0–9 scale (0 = no sign of infection and 9 = > 90% leaf and stem area infected) [4]. Six sets of PS, including five individual year datasets and the 5-year average, were used for GP modelling. The function “chart.Correlation” of the R package PerformanceAnalytics (v1.5.2, https://cran.r-project.org/web/packages/PerformanceAnalytics/index.html) was used to analyse correlations between different PS datasets and draw histograms and scatter plots.

### 4.3. Genomic Data

A total of 258,873 SNPs were obtained from the 370 accessions after pruning by removing redundant SNPs [4]. The missing data of SNPs (on average 14.13% of a missing data rate) were imputed using Beagle v.4.2 with default parameters [61]. Our previous GWAS analyses of PS in flax were conducted separately for combinations of the five individual year and the 5-year average datasets with ten statistical methods [4]. The statistical methods for GWAS included three single locus models (GLM [62], MLM [63] and GEMMA [64]) and seven multi-locus models (FarmCPU [65], mrMLM [66], FASTmrEMMA [67], ISIS EM-BLASSO [68], pLARmEB [69], pKWmEB [70], FASTmrMLM [71]). For GLM, MLM and FarmCPU, the first six principal components (PCs), accounting for 33.04% of the total variation, were chosen as covariates to measure population structure, while Frappe (http://med.stanford.edu/tanglab/software/frappe.html) was used to estimate the population structure of the 370 accessions for other six multi-locus models. GEMMA does not require a Q matrix. The threshold of significant associations for all three single-locus methods (GLM, MLM and GEMMA) and the multi-locus method FarmCPU was determined by a critical *p* value (α = 0.05) subjected to Bonferroni correction, that is, the corrected *p* value = 1.93 × 10^−7^ (0.05/258,873 SNPs), while a log of odds (LOD) score of three was used to detect robust association signals for the remaining six multi-locus models. The R package MVP (https://github.com/XiaoleiLiuBio/MVP) was used for GWAS analyses for the GLM, MLM and FarmCPU, the GEMMA software (https://github.com/genetics-statistics/GEMMA) for GEMMA and the R package mrMLM (https://cran.r-project.org/web/packages/mrMLM/index.html) for the additional six multi-locus models. The details of GWAS analyses were described in Reference [4]. A total of 500 non-redundant QTL for PS were identified from 370 diverse flax accessions, including 134 QTL that statistically stable in all five years and 67 QTL with relatively stable and large effects [4]. These three QTL datasets (500 unique QTL, 134 statistically stable QTL and 67 stable and large-effect QTL) were used for GP model construction. In addition, we performed Pearson’s χ^2^ test with Yate’s continuity correction to detect all SNPs significantly associated with PS using a 10^−5^ probability level. The three QTL sets and the genome-wide SNP set were used to construct the GP models. Thus, GP models with the 24 combinations of the four marker sets and the six phenotypic datasets were built and compared.

### 4.4. Genomic Prediction Models

Three statistical methods RR-BLUP [9,17,20], Bayesian LASSO (BL) [20,25,33] and Bayesian ridge regression (BRR) [25,72] were used to build GP models for PS. These predictive models estimate marker effects by modelling markers as random effects. No fixed effects were fitted in the models. The statistical models and their computation procedures are described in detail elsewhere [40,73]. The R package rrBLUP [56] was used to fit the RR-BLUP model and the R package BLR [74] was used to fit the BL and BRR models. The parameters used to fit BL and BRR were determined based on suggestions of de los Campos et al. [74]. Broad-sense heritability (0.25) of PS estimated in the population [3] was used. When preparing QTL marker data for model construction, the positive-effect allele of the tag SNP of a QTL was coded ‘1’ and the alternative allele ‘−1’. Similarly for the SNP marker set, the reference allele of an SNP was coded ‘1’ and the alternative allele ‘−1’. Missing data were coded ‘0’. The EM algorithm implemented in the R package rrBLUP [56] was used to impute the missing marker data because missing marker data were not allowed in the model construction.

### 4.5. Evaluation of Prediction Models

Two validation methods were used to evaluate prediction models generated from combinations of statistical models, marker sets and PS datasets. The first method was a five-fold random cross-validation. The 370 flax accessions were randomly partitioned into five subsets. For a given partition, each subset was in turn used as validation or test data and the remaining four subsets made the training dataset. This partitioning was repeated 500 times. In this manner, a total of 2500 training data sets were created to build GP models and estimate marker effects. These were used to predict the breeding values of the individuals in the corresponding 2500 test/validation datasets. The accuracy of the genomic predictions (*r*) was defined by the Pearson’s simple correlation coefficient between the genetic values predicted by GP and the observed phenotypic values. The relative efficiency of genomic prediction over phenotypic selection (*RE*) was estimated using |*r|*/$H^{2}$ [26,27], where $H^{2}$ refers to the broad-sense heritability of PS, estimated to be 0.25 [3]. *RE* was used as a criterion to compare the response to one cycle of genome-wide selection versus one cycle of phenotypic selection. Means of *r* and *RE* of the 500 samplings for each marker set, GP model and PS dataset were used to describe the prediction accuracy of GP and the efficiency of one GP cycle relative to one phenotypic selection cycle, respectively. To compare different marker and PS datasets, a joint analysis of variance with Tukey multiple pairwise-comparisons was performed to test the statistical significance of differences in *r* and *RE* using R. As a case study, we randomly selected 20% of all 370 accessions as validation dataset and used the remaining 277 accessions as training dataset to build a GP model for genomic prediction of unknown germplasm.

The second cross-validation approach involved comparisons across different PS datasets, that is, each of the six complete PS phenotypic datasets were used as training datasets to build GP models that were applied to itself and to the other five phenotypic datasets. The same set of markers for all 370 accessions was used for training and validation. This method tests the relevance of models built based on single year phenotypic data to predict phenotypes measured in different years. 

### 4.6. Phenotypic Variation Explained by Markers

The phenotypic variation explained by all markers in various marker sets, denoted $h_{SNP}^{2}$, was estimated for all PS datasets based on the mixed linear model [75] implemented in the GCTA software [76]. The detailed calculation is described in Reference [77].

## 5. Conclusions

Using a diverse worldwide flax core collection of 370 accessions as a training and test population with 500 QTL identified by GWAS, the 5-year average PS data and the RR-BLUP statistical model, we developed a highly effective GP model with a prediction accuracy as high as 0.92 for pasmo, a low heritability and high inheritance complexity trait. This is the highest reported accuracy value of all GP models for plant disease resistance traits and comparable with previously published results. As an alternative, we developed a linear regression prediction model based on NPQTL that also produced a high prediction accuracy of 0.86. The GP model and the NPQTL-based regression equation were validated and deemed to be applicable to the evaluation of flax germplasm including parent selection for PR. The use of all potential QTL associated with a target trait would be beneficial because the exclusion of a large proportion of unrelated markers would facilitate the construction of highly accurate GP models.

## Figures and Tables

**Figure 1 ijms-20-00359-f001:**
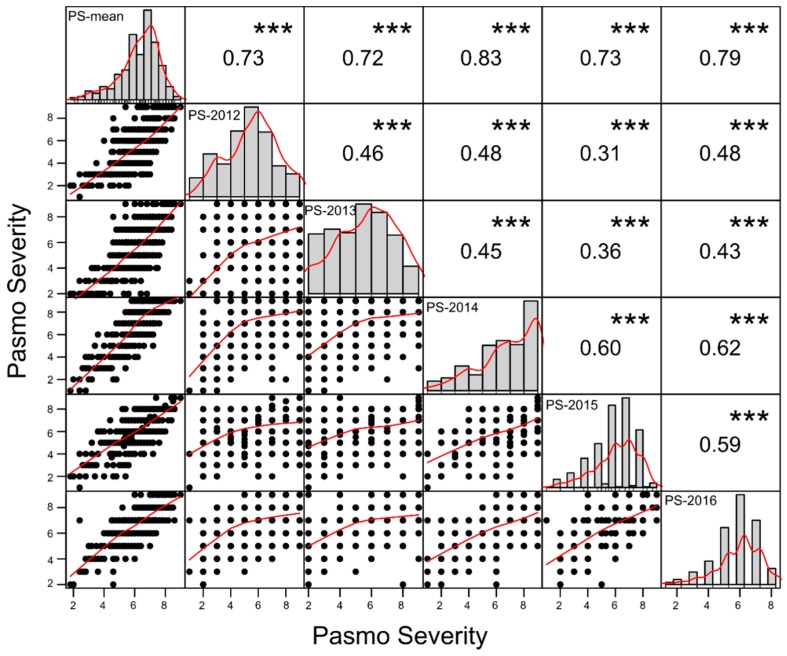
Dot plots (lower triangle), histograms (diagonal) and Pearson correlations (upper triangle) between six pasmo severity datasets. Best curves are fitted in dot plots and histograms. *** represents significance at the <0.001 probability level.

**Figure 2 ijms-20-00359-f002:**
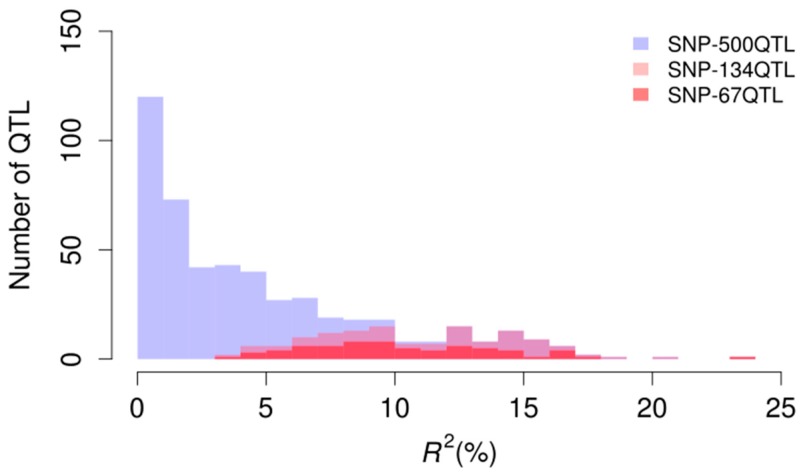
Distribution of *R*^2^ (%) (phenotypic variation explained by individual QTL) in the three QTL marker sets.

**Figure 3 ijms-20-00359-f003:**
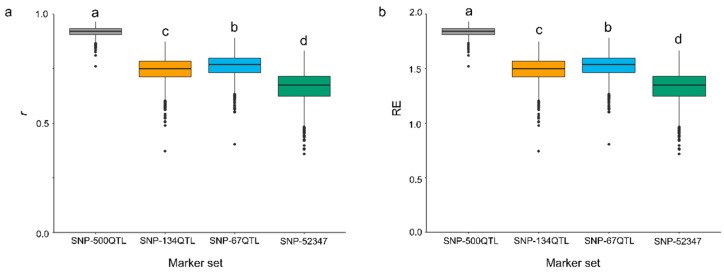
Accuracy (*r*) (**a**) and relative efficiency (*RE*) (**b**) of RR-BLUP prediction models built with combinations of four marker sets using the five-year average PS dataset (PS-mean) and random five-fold cross-validations. Letters above box plots indicated statistical significance (*p* < 0.05) for *r* and *RE* among marker sets.

**Figure 4 ijms-20-00359-f004:**
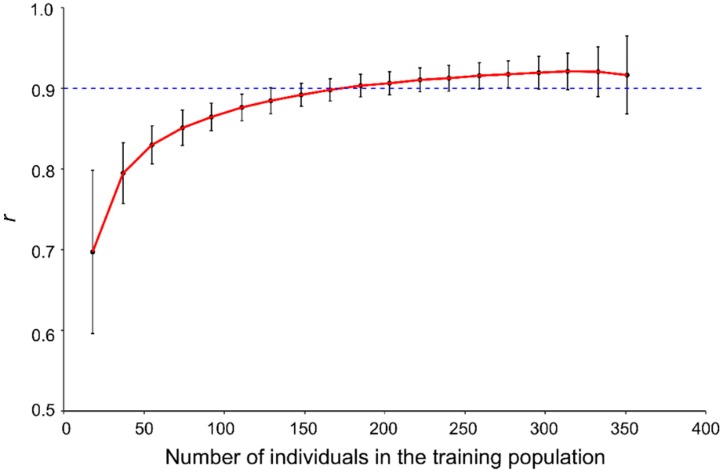
Relationship between the genomic prediction accuracy (*r*) and the size of the training population based on the SNP-500QTL marker set, the PS-mean dataset and the RR-BLUP models. The dash line represents a prediction accuracy of 0.9.

**Figure 5 ijms-20-00359-f005:**
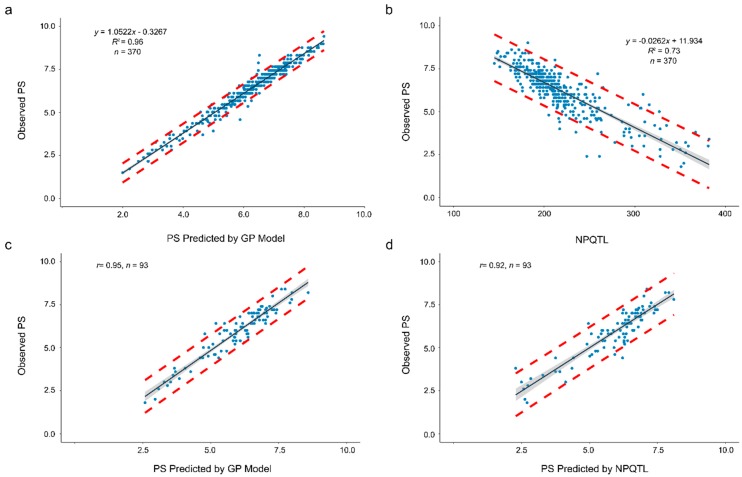
Relationship of observed pasmo severity (PS) with PS predicted by a GP model (**a**,**c**) or with PS predicted by the number of QTL with positive-effect alleles (NPQTL) (**b**,**d**). (**a**) Linear regression of observed PS (*y*) to predicted PS (*x*) using the genomic prediction model built with the PS-mean dataset and the SNP-500QTL marker set of all 370 accessions as training data set. (**b**) Linear regression of observed PS (*y*) to NPQTL (*x*) in the 370 flax accessions. (**c**) Relationship of observed PS of 93 randomly chosen accessions with the PS predicted by the genomic model constructed with the SNP-500QTL marker set and PS-mean dataset when a random subset of 277 accessions was used as training population. (**d**) Relationship of observed PS of 93 randomly chosen accessions with the PS predicted by NPQTL (Appendix A) The red dashed lines represent upper and lower boundaries of the 95% prediction intervals, that is, it is expected that the value of a sample lies within that prediction interval in 95% of the samples. The grey band represents the 95% confidence interval, that is, 95% of those intervals include the true value of the population mean.

**Table 1 ijms-20-00359-t001:** Pasmo severity of 370 flax accessions across five years in the field condition.

Data Set	$\bar{x}\;\pm$ *s*	Range	*CV* (%)
PS-2012	5.57 ± 1.86	1.00–9.00	32.76
PS-2013	5.69 ± 1.91	2.00–9.00	33.20
PS-2014	6.86 ± 2.07	1.00–9.00	29.41
PS-2015	6.11 ± 1.55	1.00–9.00	25.44
PS-2016	6.72 ± 1.37	2.00–9.00	20.39
PS-mean	6.22 ± 1.32	1.80–9.00	21.27

$\bar{x}$: average pasmo severity across five years; *s*: standard deviation; *CV*: coefficient of variation.

**Table 2 ijms-20-00359-t002:** Phenotypic variation of pasmo severity (PS) ($h^{2}$ ± *s*) explained by the four marker sets.

PS Dataset	Marker Set			
	SNP-500QTL	SNP-134QTL	SNP-67QTL	SNP-52347
PS-mean	0.72 ± 0.04	0.27 ± 0.05	0.29 ± 0.05	0.54 ± 0.07
PS-2012	0.64 ± 0.06	0.18 ± 0.05	0.16 ± 0.04	0.43 ± 0.08
PS-2013	0.63 ± 0.06	0.12 ± 0.04	0.12 ± 0.04	0.38 ± 0.08
PS-2014	0.65 ± 0.06	0.23 ± 0.05	0.20 ± 0.05	0.45 ± 0.08
PS-2015	0.56 ± 0.06	0.20 ± 0.05	0.17 ± 0.04	0.44 ± 0.09
PS-2016	0.53 ± 0.06	0.18 ± 0.05	0.18 ± 0.05	0.38 ± 0.07

**Table 3 ijms-20-00359-t003:** Accuracy (*r*) and relative efficiency (*RE*) values of the 24 combinations representing the four marker sets and six pasmo severity (PS) datasets using RR-BLUP obtained using a random five-fold cross-validation.

Marker Set	PS Dataset	$r_{\;}(\bar{x}\pm$*s*) ^1^	$\mathrm{RE}\;(\bar{x}\pm$*s*) ^1^
SNP-500QTL	PS-mean	0.92 ± 0.02a	1.84 ± 0.04a
	PS-2012	0.84 ± 0.03b	1.68 ± 0.06b
	PS-2013	0.81 ± 0.04c	1.62 ± 0.07c
	PS-2014	0.82 ± 0.04c	1.63 ± 0.07c
	PS-2015	0.76 ± 0.05d	1.52 ± 0.09d
	PS-2016	0.76 ± 0.05d	1.52 ± 0.11d
SNP-134QTL	PS-mean	0.75 ± 0.06e	1.49 ± 0.11e
	PS-2012	0.68 ± 0.06f	1.36 ± 0.11f
	PS-2013	0.60 ± 0.07ij	1.19 ± 0.14ij
	PS-2014	0.60 ± 0.07i	1.21 ± 0.14i
	PS-2015	0.47 ± 0.09o	0.94 ± 0.18o
	PS-2016	0.56 ± 0.09l	1.12 ± 0.17l
SNP-67QTL	PS-mean	0.76 ± 0.05d	1.53 ± 0.1d
	PS-2012	0.67 ± 0.06g	1.35 ± 0.11g
	PS-2013	0.60 ± 0.07ij	1.20 ± 0.14ij
	PS-2014	0.60 ± 0.07ij	1.20 ± 0.14ij
	PS-2015	0.50 ± 0.09n	1.00 ± 0.17n
	PS-2016	0.59 ± 0.08k	1.17 ± 0.17k
SNP-52347	PS-mean	0.67 ± 0.07g	1.33 ± 0.14g
	PS-2012	0.63 ± 0.06h	1.27 ± 0.12h
	PS-2013	0.59 ± 0.07jk	1.19 ± 0.14jk
	PS-2014	0.53 ± 0.08m	1.06 ± 0.17m
	PS-2015	0.38 ± 0.09q	0.77 ± 0.17q
	PS-2016	0.46 ± 0.09p	0.93 ± 0.18p

^1^ Different letters represent multiple test significance among the 24 combinations at the 0.05 probability level.

**Table 4 ijms-20-00359-t004:** Accuracy (*r*) and relative efficiency (*RE*) of genomic prediction for pasmo severity in different years using the RR-BLUP model built with the SNP-500QTL marker set and the PS-mean phenotypic data using all 370 accessions as training data set.

PS Dataset for Prediction	*r*	*RE*
PS-mean	0.98	1.96
PS-2012	0.73	1.46
PS-2013	0.71	1.42
PS-2014	0.81	1.62
PS-2015	0.71	1.43
PS-2016	0.77	1.55

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
