# Peer review of "Evaluation of Genomic Prediction for Pasmo Resistance in Flax"

_ijms, 2019, doi:10.3390/ijms20020359_

Reviewer 1 Report

For low heritability traits, a challenged task is how to improve the accuracy of genomic prediction. In this manuscript, the authors compared the efficiencies of four marker sets. As a result, the 500 QTLs with the 5-year pasmo severity average as the training dataset had the highest GP accuracy (0.92) in five-fold random cross-validation experiments. Clearly, the results are good. The result can be considered for publication in this Journal.

Major revision

1.     Lines 101-103, the authors should describe how to obtain the 500, 134, and 67 previously identified QTL. As we know, these QTL should be identified jointly from several single- and multi-locus GWAS methods. Or users cannot obtain similar results in other crops and datasets. I feel that the authors may add one sub-section to describe the related information in Materials and Methods, including the related single- and multi-locus GWAS methods.

2.      The authors pointed out that RR-BLUP is the best method for genomic prediction. Actually, it is for the traits with many small-effect polygenes. If major genes exist, GBLUP may be the best one. The reasons are as follows. RR-BLUP assumes markers to have equal variance while the GBLUP lets various markers to have different variances. Please explain your results using the above reason.

3.      In genomic prediction, the multi-year trait average or its BLUP values are frequently adopted. Please discuss how to select the two kinds of phenotypes.

4.      The purpose of genomic prediction is to obtain the best individual in crop breeding. Please address this issue.

Minor revision

1.      Line 84, “The Bayesian LASSO (BL) assumes markers to have equal variances” is wrong. This is because RR-BLUP assumes markers to have equal variance while the Bayesian LASSO lets various markers to have different variances.

2.      Lines 86-87, “at most n variables” should be “at most n−1 variables”.

3.     In Table 2, “Different letters represent multiple test significance”. However, there is no information for multiple testing method. The same situation was found in Figure 2.

4.     In Figure 2, accuracy (r) and relative efficiency (RE) aren’t clear. Please explain them. Actually, r was the Pearson's simple correlation coefficient between the predicted and observed phenotypic values, and RE was r/H2, where H2 was the broad-sense heritability. In addition, if relative efficiency is defined by r2/H2, its biological meaning is clearer and its values are between 0 and one.

5.     Figures 4 to 6 are similar, and can be integrated into one figure.

Author Response

Dear the anonymous reviewer and editors,

Thank you for the positive comments and all constructive suggestions. We have made extensive revisions and English editing accordingly. All majorly revised sections are highlighted in yellow.

The point-by-point comments and answers are listed below. Thank you for your further review and comments, and consideration for publication of this manuscript in IJMS.

Frank You

PS. Comments and Answers

 For low heritability traits, a challenged task is how to improve the accuracy of genomic prediction. In this manuscript, the authors compared the efficiencies of four marker sets. As a result, the 500 QTLs with the 5-year pasmo severity average as the training dataset had the highest GP accuracy (0.92) in five-fold random cross-validation experiments. Clearly, the results are good. The result can be considered for publication in this Journal.

Major revision

1.      Lines 101-103, the authors should describe how to obtain the 500, 134, and 67 previously identified QTL. As we know, these QTL should be identified jointly from several single- and multi-locus GWAS methods. Or users cannot obtain similar results in other crops and datasets. I feel that the authors may add one sub-section to describe the related information in Materials and Methods, including the related single- and multi-locus GWAS methods.

Answer:

Thanks for the suggestion. We agree that our good results are primarily resulted from using the large and minor effect QTL identified by single- and multi-locus statistical methods as a training marker data set. We have revised the paragraph about QTL identification under “4.3 Genomic Data” in M & M (Lines 399-426). 

2.      The authors pointed out that RR-BLUP is the best method for genomic prediction. Actually, it is for the traits with many small-effect polygenes. If major genes exist, GBLUP may be the best one. The reasons are as follows. RR-BLUP assumes markers to have equal variance while the GBLUP lets various markers to have different variances. Please explain your results using the above reason.

Answer:

Thanks for the suggestion. Because GBLUP is theoretically equivalent to RR-BLUP that assumes markers to have equal variance. We thought the reviewer meant some models that have specific variance for each marker and are able to do variable selection in the models such as Bayesian B, which mostly outperforms RR-BLUP if a trait has only a few major genes. We revised the Discussion section “3.3 Accuracy of GP Modeling by Environment, Training Population, and Statistical Methods” (Lines 322-345).

3.      In genomic prediction, the multi-year trait average or its BLUP values are frequently adopted. Please discuss how to select the two kinds of phenotypes.

Answer:

Thanks for the suggestion. In this study, we adopted a type-2 modified augmented design (MAD2) (Lin and Poushinsky, 1985; You et al. 2013) to evaluate a large number of genotypes. In this design, all test genotypes have only one replication but two main control varieties and three sub-control cultivars have multiple replications that are used to estimate error variance. Phenotypes in each year are adjusted according to control varieties. All these data analyses have been described in You et al. (2013). Based on the adjusted phenotypes of multiple years (without replications in each year), we noticed BLUEs and BLUPs of genotypes were equivalent to the averages over years. Thus we used the term “average over multiple years” for GWAS and genomic prediction. Because of equivalence between BLUES, BLUPs and means in our case, we suggest no further discussion for the selection of trait average and BLUP in this manuscript, but we added a sentence to indicate the equivalence between BLUPs and the means (Lines 296-298).

4.      The purpose of genomic prediction is to obtain the best individual in crop breeding. Please address this issue.

Answer:

Thank you for the suggestion. We added one sub-section “3.4 Breeding Application of Genomic Prediction” to discuss the application of genomic prediction in breeding (Lines 356-372).

Minor revision

1.      Line 84, “The Bayesian LASSO (BL) assumes markers to have equal variances” is wrong. This is because RR-BLUP assumes markers to have equal variance while the Bayesian LASSO lets various markers to have different variances.

Answer:

Thanks for pointing out this mistake. We have corrected it (Lines 84-85). Like the Bayes B method, the Bayesian LASSO assume a different variance for each marker.

2.      Lines 86-87, “at most n variables” should be “at most n−1 variables”.

Answer:

Corrected. Thanks!

3.      In Table 2, “Different letters represent multiple test significance”. However, there is no information for multiple testing method. The same situation was found in Figure 2.

Answer:

Thanks for the comments. Tukey multiple pairwise-comparisons were used in the study. A sentence about the method in M & M were added accordingly (Line 458).

4.      In Figure 2, accuracy (r) and relative efficiency (RE) aren’t clear. Please explain them. Actually, r was the Pearson's simple correlation coefficient between the predicted and observed phenotypic values, and RE was r/H2, where H2 was the broad-sense heritability. In addition, if relative efficiency is defined by r2/H2, its biological meaning is clearer and its values are between 0 and one.

Answer:

Thanks for the comments. The definition of r and RE are defined and described in Lines 449-455. We corrected a minor error for RE definition to |r|/H2. The meaning of RE was explained in Lines 451-453. Thank you for suggestion for the suggestion of r2/H2. Our definition of RE is based on Professor Rex Burnardo (U of Minnesota) (Ziyomo and Bernardo 2013 and personal communication to Prof. Bernardo). Here are its theoretical bases:

References:

Ziyomo, C.; Bernardo, R. Drought tolerance in maize: indirect selection through secondary traits versus genomewide selection. Crop Sci. 2013, 53, 1269-1275.

Dekkers, J.C. Prediction of response to marker-assisted and genomic selection using selection index theory. J. Anim. Breed. Genet. 2007, 124, 331-341.

As Dekkers (2007) indicated, path coefficient analysis gives the correlation between marker-predicted values (M) and genotypic values (G) as

rMG = rMP/h

where rMP is the observed correlation between M and phenotypic values (P), and h is the square root of the estimate of heritability for the trait. In other words, given that the correlation for the M --> G path is rMG and the correlation for the G --> P path is rGP = h, the correlation for the M --> G --> P path is rMP = rMG h.

The ratio between the response to genome-wide selection and the response to phenotypic selection is derived from classical indirect-selection theory. The relative efficiency of selecting for a secondary trait (genome-wide markers) to improve a primary trait, versus direct (phenotypic) selection for the primary trait itself as:

RE = (Absolute value of genetic correlation between the two traits) (h for secondary trait) / (h for primary trait)

In this context, the genetic correlation is rMG; the h of marker data is assumed as h2 = h = 1 (i.e., no errors in marker data), and the square root of the heritability of the trait is h. The RE then becomes

RE = | rMG | / h      = |rMP / h | / h      = |rMP| / h2

So indeed, RE can be used as a criterion to compare the response to one cycle of genome-wide selection versus one cycle of phenotypic selection. In fact, rMP/h2 can be well less than 1 but still more efficient than phenotypic selection if three generations of genome-wide selection can be done per year.

To avoid this long explanations, we just cited the previous publications in the manuscript.

5.      Figures 4 to 6 are similar, and can be integrated into one figure.

Answer:

Thanks for suggestions. Figures 4 to 6 have been merged into one figure (Lines 215-228).

Reviewer 2 Report

Manuscript (ijms-406898) “Evaluation of Genomic Prediction for Pasmo Resistance in Flax” by Zhang et al. presents an interesting study about the high-throughput sequencing of the Tree peonytranscriptome in seed.

However despite the large amount of work done and data presented, the methodology and results presents some deficiencies. Mainly, authors must better explain the phenotypic characterization of assayed material.

For these reasons, this manuscript is acceptable for publication in International Journal of Molecular Sciences AFTER A MODERATE/MAJOR REVISION.

The major points for the revision of the manuscript are:

Objective of the work must be clearly specified in a separated paragraph excluding bibliography.

Plant material assayed must be better explained. In my opinion a clear phenotypic characterization is necessary of the two assayed species.

In this sense in the results part a new section with the results of plasmo resistance data is required.

This phenotypic characterization should be included

Author Response

Dear the anonymous reviewer and editors,

Thank you for the positive comments and all constructive suggestions. We have made extensive revisions and English editing accordingly. All majorly revised sections are highlighted in yellow.

The point-by-point comments and answers are listed below. Thank you for your further review and comments, and consideration for publication of this manuscript in IJMS.

Frank You

PS. Comments and Answers

Manuscript (ijms-406898) “Evaluation of Genomic Prediction for Pasmo Resistance in Flax” by Zhang et al. presents an interesting study about the high-throughput sequencing of the Tree peonytranscriptome in seed.

However despite the large amount of work done and data presented, the methodology and results presents some deficiencies. Mainly, authors must better explain the phenotypic characterization of assayed material.

For these reasons, this manuscript is acceptable for publication in International Journal of Molecular Sciences AFTER A MODERATE/MAJOR REVISION.

Answer:

Thank the anonymous reviewer for the positive comments and constructive suggestions. We have made extensive revisions accordingly and all revised sections have been highlighted in yellow.

The major points for the revision of the manuscript are:

Objective of the work must be clearly specified in a separated paragraph excluding bibliography.

Answer:

Thanks for the suggestion. We have revised and re-organized a seperate paragraph for the objectives of this research (Lines 104-107).

Plant material assayed must be better explained. In my opinion a clear phenotypic characterization is necessary of the two assayed species.

Answer:

Thanks for the suggestions. We have revised the section “4.2. Pasmo Resistance Data” to include some detailed phenotypic data and their analysis (Lines 380-396).

In this sense in the results part a new section with the results of plasmo resistance data is required.

This phenotypic characterization should be included.

Answer:

Thanks for the suggestion. We have added a new section “2.1. Evaluation of Pasmo Resistance” with a new table (Table 1) and a new figure (Figure 1) (Lines 109-121 and Lines 136-140). Re-numbering of other tables and figures were made accordingly.

Round  2

Reviewer 2 Report

Authors have revised correctly their manuscript.